# Validation of EMI-2 Radiometric Performance with TROPOMI over Dome C Site in Antarctica

Jingming Su [1,2,3], Fuqi Si [1,*], Minjie Zhao [1], Haijin Zhou [1] and Yan Hong [3]

1 Key Laboratory of Environmental Optics and Technology, Anhui Institute of Optics and Fine Mechanics, Chinese Academy of Sciences, Hefei 230031, China

2 University of Science and Technology of China, Hefei 230026, China

3 College of Electrical and Information Engineering, Anhui University of Science and Technology, Huainan 232001, China

* Correspondence: sifuqi@aiofm.ac.cn

**Abstract:** (1) The Environmental Trace Gases Monitoring Instrument-2(EMI-2) is a high-quality spaceborne imaging spectrometer that launched in September 2021. To evaluate its radiometric calibration performance in-flight, the UV2 and VIS1 bands of EMI-2 were cross-calibrated by the corresponding bands (band3 and band4) of TROPOMI over the pseudo-invariant calibration site Dome C. (2) After angle limitation and cloud filtering of the Earth radiance data measured by EMI-2 and TROPOMI over Dome C, the top of atmosphere (TOA) reflectance time series were calculated. The spectral adjustment factors (SAF) were derived from the solar spectrum measured by the sensor to minimize the uncertainties caused by the different spectral response functions (SRF) of sensors. In addition, a correction method based on the radiative transfer model (RTM) SCIATRAN was used to suppress unaccounted angular dependence of atmospheric scattering. The radiation performance of EMI-2 is evaluated using the TOA reflectance ratio of EMI-2 and TROPOMI, combining the SAF correction and RTM-based correction methods. (3) It was shown that the time series trending of the TOA reflectance ratio between EMI-2 measurements and TROPOMI demonstrate flat characteristics and strong correlation. The mean reflectance ratios range from 0.998 to 1.09. The standard deviation of the reflection ratio is less than 3%. For 328 nm, 335 nm, 340 nm, 460 nm, and 490 nm, the mean values are close to one, and the relative radiometric bias estimated through EMI-2 and TROPOMI intercalibration is less than 3%, and for other wavelengths, the biases are less than 6%, except for 416 nm, which behaves higher than 7%. The cross-calibration results show that the radiometric calibration of EMI-2 is within the relative accuracy requirement.

**Keywords:** radiometric calibration; TOA reflectance; cross-calibration; SCIATRAN; EMI-2; TROPOMI

## 1. Introduction

The self-built EMI-2, a high-quality four-channel spaceborne imaging spectrometer, is the successor of EMI with a wavelength range from 240 nm to 710 nm, which was launched on the GaoFen-5(02) satellite in September 2021. Many trace gas products, such as $NO_2$, $O_3$, and $SO_2$, as well as cloud and aerosol properties, can be retrieved with spectral coverage and resolution. The quality of the retrieved properties depends on the quality of the calibration of the satellite instrument. For trace gas products retrieved by differential optical absorption spectroscopy (DOAS), the absolute calibration of the reflectance is not so valuable because the radiation errors of any constant in the DOAS method can be canceled out [1]. However, the retrieval codes of cloud and aerosol products depend on accurate radiometric calibration [2–4]. To ensure the high quality of in-orbit radiometric calibration of EMI-2, a series of laboratory calibrations and instrument characterizations before launch have been conducted, and the calibration uncertainty is about 5% [5].

After launch, due to the launch-related stress, the space environment, and the aging of sensors, the remote sensing instruments may degrade over time. Thus, it is meaningful

to monitor the calibration stability and evaluate the performance and reliability of the radiometric calibration continuously.

EMI-2 has Earth, Sun, and white light source (WLS) observation ports. Figure 1 shows the optical layout. The Sun and Earth have different optical paths. In Solar mode, solar irradiance enters the telescope via an onboard diffuser and folding and secondary mirrors. In Earth mode, atmospheric scattering light enters the telescope through a primary and secondary mirror. Monitoring measurements are scheduled on a regular basis to be able to correct degradation and drift effects [5].

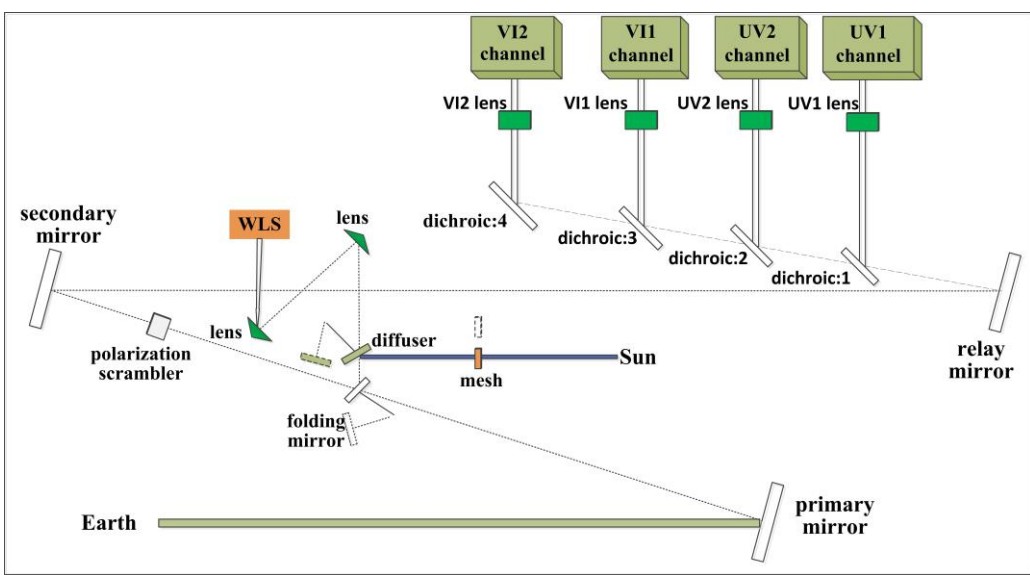

**Figure 1.** EMI-2 optical layout.

As soon as EMI-2 entered orbit, it began to conduct daily detection and monitoring based on solar irradiance measurements. The measured variation of solar irradiance can be used as a reference for the instrument decay parameter. However, due to the slightly different light path between the solar irradiance and the Earth's radiance, the change in solar irradiance cannot exactly represent the degradation of the Earth's radiance [6]. A white light source (WLS) equipped on EMI-2 can only provide information on part of the internal optical path used for radiance and solar irradiance measurements. Therefore, the radiance data observed from the Earth may not be perfectly corrected. Assuming that the Sun is a stable light source, the TOA reflectance is determined by the ratio of solar irradiance and Earth radiance measurements, which can eliminate instrumental characteristics. Thus, the TOA reflectance datasets are selected for radiometric calibration of the EMI-2 instrument in flight.

In fact, the in-flight radiometric calibration of the reflectance has been implemented in all kinds of ways. Lieuwe G. compared the reflectance results measured by TROPOMI with that of the simulation obtained by the radiative transfer code DAK [7]. Jaross proposed comparing remotely-sensed radiances to predictions originating from a radiative transfer model [8]. Xiong made use of an inner set of onboard calibrators and observations over Dome C to evaluate the stability and consistency of MODIS [9]. Clark J. introduced an empirical approach rather than a radiative transfer model to eliminate the solar zenith angle dependence from the intensities to realize the intercalibration of nine UV sensing instruments over Antarctica and Greenland [10]. These methods can effectively monitor the radiometric stability and absolute radiometric accuracy of sensors to some extent, but their absolute radiometric accuracy highly relies on the stability of their own calibrators.

The cross-calibration and intercomparison among sensors is another effective way. Jing assessed the radiometric performance of multiple VNIR bands of the GOES-16 ABI over the Sonoran Desert by intercomparison measurements between the Suomi National Polar-Orbiting Partnership (S-NPP) and NOAA-20 Visible Infrared Imaging Radiometer

Suite [11]. Tang accomplished radiometric cross-calibration of the Ziyuan-3 Satellite with GF1 satellite and Landsat-8 [12]. Gao used the radiometric cross-calibration method to monitor the degradation of GF-4/VNIR with the aid of Landsat8/OLI, Sentinel-2/MSI, and Terra/MODIS [13]. These research findings indicate that the radiometric cross-calibration scheme can be reliable and effective. For EMI series sensors, long-term irradiance monitoring and pseudo-constant calibration site monitoring are often used [14,15].

In this research, the in-flight cross-calibration method is adopted to evaluate the validation of EMI-2 radiometric performance. Section 2 provides a brief description of the Dome C site and the instruments. Section 3 describes the methodology of data selection and the RTM-based cross-calibration method. Section 4 presents the results and discussion about the radiometric calibration of EMI-2, and Section 5 gives the conclusion.

## 2. Site Selection and Cross-Calibration Sensor Selection

### 2.1. Site Selection

The pseudo-invariant calibration site Dome C (75.1°S, 123.4°E), shown in Figure 2, is located in the Eastern Antarctic Plateau. It features a high altitude (~3200 m), uniform land surfaces with snow and ice, and a slight surface slope, which shows a negligible effect on BRDF and viewing reflectance uniformity. The stable atmospheric conditions feature as cold, dry with consistently less cloud cover, and least affected by cloud cover [16]. The location is far from the coast (>1000 km), and the atmospheric aerosol and water vapor content are low. These factors can reduce the radiance uncertainty to the minimum (less than ±2%) [9]. All things indicate that Dome C is very suitable as a calibration site for the ultraviolet and visible channels observation.

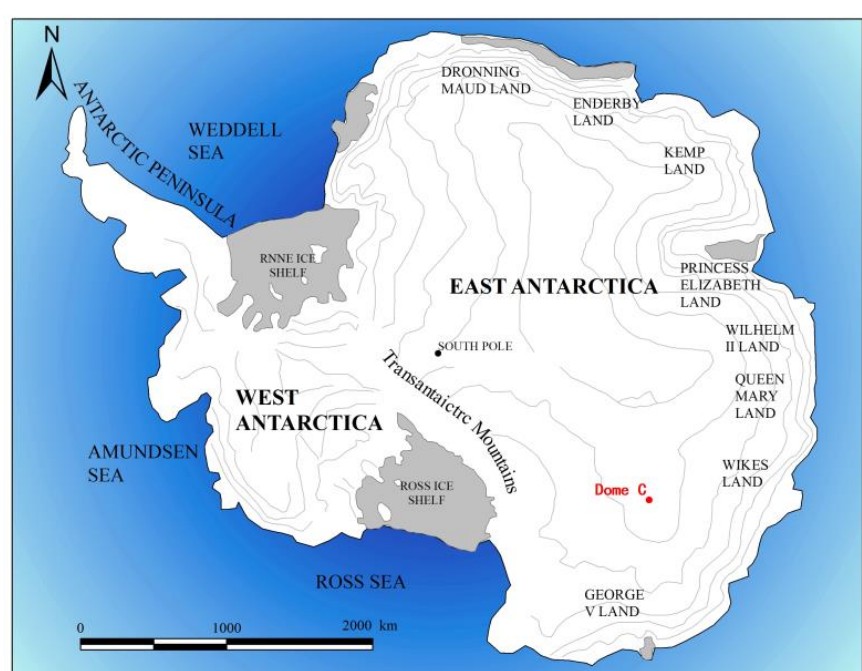

**Figure 2.** Geographical location of the calibration site.

### 2.2. EMI-2 and TROPOMI

EMI-2 is a nadir-viewing imaging spectrometer with four spectral detectors onboard Gaofen-5(02), which uses the differential optical absorption spectroscopy (DOAS) technique to retrieve trace gases. It has four spectral channels: ultraviolet channel 1 (UV1, 240–311 nm), ultraviolet channel 2 (UV2, 311–401 nm), visible channel 1 (VIS1, 401–550 nm), and visible channel 2 (VIS2, 545–710 nm). The imaging system enables daily global coverage via a push-broom configuration from a Sun-synchronous orbit at 705 km, with spatial resolution as low as $7 \times 16$ km$^2$. Its ascending node equatorial crossing time is at 13:30.

The TROPOMI instrument onboard the Sentinel-5 Precursor (S5P) satellite was launched in October 2017 and configured with four detectors; each has two spectral bands [17]. The wavelength range has UV (267–499 nm), near-infrared (661–786 nm), and short-wave infrared (2300–2389 nm). The satellite operates in a near-polar Sun-synchronous orbit with an average altitude of 824 km above the Earth's surface [18]. The time of crossing the equator is 13:30 LT, and the repetition period is 17 d. The resolution of TROPOMI has been 5.6 km × 3.6 km since 6 August 2019. The detailed parameters comparison between EMI-2 and TROPOMI is shown in Table 1.

**Table 1.** The comparison of parameters of EMI-2 and TROPOMI.

| Parameter | EMI-2 | TROPOMI |
|---|---|---|
| Spectral range | UV1:240–311 nm; UV2:311–401 nm; VIS1:401–550 nm; VIS2:545–710 nm; | BD1:267–299 nm; BD2:299–331 nm; BD3:304–400 nm; BD4:400–498 nm; BD5:661–724 nm; |
| Spectral resolution | 0.3~0.6 nm | 0.5 nm |
| Total field of view | 114° (2600 km × 6.5 km ground) | 114° (2600 km × 3.6 km ground) |
| CCD detector | UV: 1072 × 1032 Pixels VIS: 1286 × 576 Pixels | UV: 1072 × 1032 Pixels VIS: 1286 × 576 Pixels |
| Spatial resolution | 13 km (Flight direction) × 24 km (Swath direction) | 5.6 km (Flight direction) × 3.6 km (Swath direction) |
| Orbit | polar, Sun-synchronous, descending node equator crossing time: 10:30 | Near-polar, Sun-synchronous, ascending node equator crossing time: 13:30 |
| Altitude | 705 km | 824 km |

The GaoFen-5(02) and S5P are all polar orbit satellites. Due to homogenous detection algorithm and similar measurement data, the TROPOMI is selected to carry out the cross-calibration evaluation on EMI-2. As shown in Table 1, the spatial resolution of TROPOMI is higher than that of EMI-2. The TROPOMI calibration has been validated in many ways, and the absolute radiometric calibration of the Earth port was good and further improved to 1.0% to 1.9% by a post-calibration of the employed external diffuser [7,17,18], which indicates that the TROPOMI is one of the best choices for carrying out cross-calibration with EMI-2. It is worth noticing that the operation orbits of these two sensors are not exactly the same, and the time of passing the Dome C site is different every day. However, due to the stable surface reflectance in the Antarctic, the differences can be ignored in the cross-calibration process.

## 3. Methods

A flowchart of the radiometric calibration validation for UV2 and VI1 channels of EMI-2 is shown in Figure 3. Firstly, the radiance measurements of EMI-2 and TROPOMI are selected according to the angular limitation requirement, and the cloud contamination is screened out simultaneously. Then, calculate the SAF factor to remove the calibration error caused by the instrumental difference, and use the RTM-based method to perform geometric effect correction of the measurement. Finally, the radiation bias of EMI2-TROPOMI is evaluated by combining the RTM correction method with SAF correction technique.

### 3.1. Data Selection

Dome C is located in a high-latitude area near the South Pole. Matching the geometry of the two sensors is important because of the high latitude of the calibration site, the large viewing angles, and the high zenith angles, creating a high TOA reflectance sensitivity to the bidirectional reflection distribution function (BRDF) and Rayleigh effect [19].

#### 3.1.1. Angular Limitation

A large diurnal variation in reflectance due to bidirectional effects is a major defect at the Dome C site for sensor calibration. The cross-calibration needs the highest-quality

observations and enough data points. The rules for data selection of EMI-2 and TROPOMI are set as follows:

(1) The study area should be centered at the Dome C site (75.1°S, 123.4°E), within a radius of 15 km (≤0.5°). If multiple pixels qualify, only the pixel with the smallest separation from the reference position is used [20].

(2) Due to the potential BRDF impact and the different observation time and geometry shapes on the evaluation of radiation bias, the observation angle is required to be less than 15°.

(3) The sun zenith angle of the South Pole is relatively large. In order to eliminate the influence of BRDF and stray light effect, the relative deviation between the azimuth of the Sun is limited to 50° to 65°.

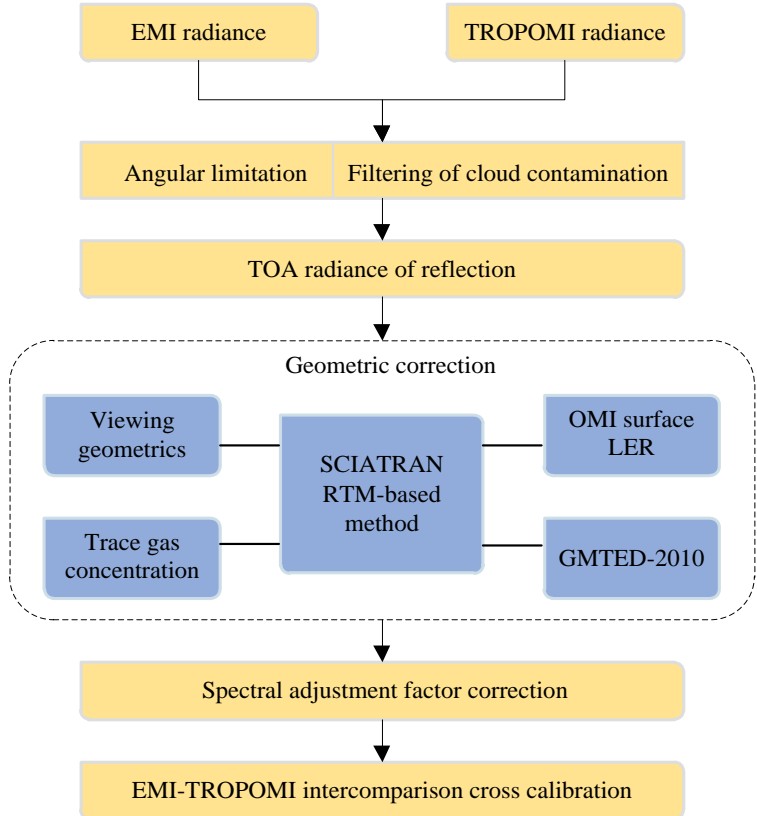

**Figure 3.** Flowchart for cross-calibration of EMI-2 to TROPOMI based on RTM method.

3.1.2. Cloud/Shadow Contamination Filtering

The histogram threshold of the reflectance method was used to remove the outlier observations contaminated by clouds or shadows. Cloud-contaminated data are removed according to the spatial uniformity (ratio of 1 standard deviation to mean reflectance over the study area) with a threshold value of 5% [20–22].

Figure 4 shows the reflectance histogram of EMI-2 before and after anomaly removal. The statistical characteristics of the data region in the south polar regions show a narrow Gaussian distribution in the selected sample region, indicating stable surface reflectance. The purple represents the original reflectivity data distribution, and the red part data means distribution after the removal of outliers, respectively. Apparently, the bin width (colored with red) has been well refined after the anomaly removal. For EMI-2 and TROPOMI observation, about 6% and 5% of the data are screened out as high cloud contamination, respectively.

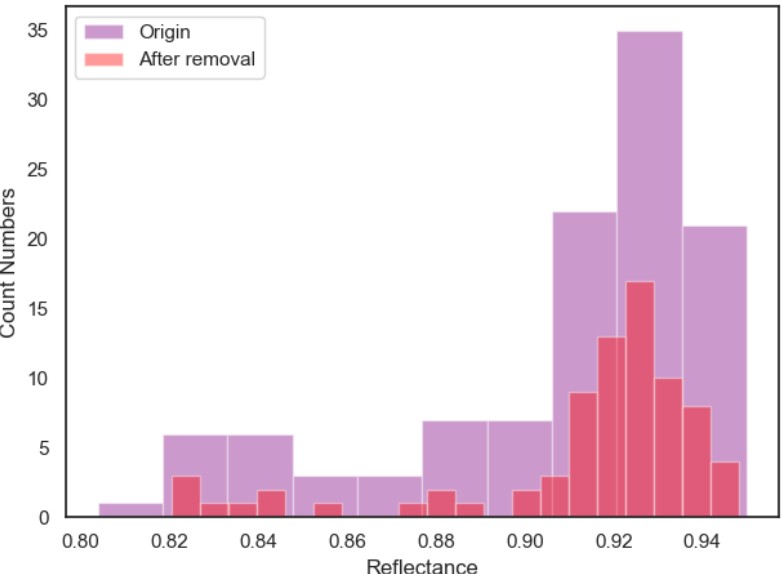

**Figure 4.** Reflectance histograms of EMI-2 UV2 (340 nm) with and without anomaly removal.

After that, the recursive filtering method is introduced to further delete the abnormal measurements possibly due to cloud/shadow contamination [11]. Based on the characteristics of high latitude and large solar zenith angle in the south polar regions, a quadratic polynomial BRDF model of solar zenith angle and surface reflectance is established. The relationship between surface reflectance and the cosine of the solar zenith angle is shown in Equation (1).

$$\rho_{\text{sensor}} = k_0 + k_1 \cos \theta_s + k_2 \cos^2 \theta_s, \tag{1}$$

where $\rho_{\text{sensor}}$ is the TOA reflectance predicted by the BRDF model, $k_0$, $k_1$, and $k_2$ are the fitting coefficients of the quadratic polynomial model, and $\theta_s$ is the solar zenith angle.

According to the polynomial fitting regression mechanism, the data with the largest left-term residual are identified as pollution observation data and need to be removed from the dataset. Repeat this operation until the fitting residual is less than 0.01.

Figure 5 shows the polynomial relationship between the TOA reflectance and SZA(cos $\theta_s$) for EMI-2/TROPOMI. On the left, the four figures represent the results before filtering, while on the right side, the four figures represent the results after filtering. The data of 340 nm of the UV2 band and 460 nm of the VI1 band are selected for comparison. It can be seen that most of the outliers in the left figures are significantly removed, as shown on the right side correspondingly. Through this recursive filtering method, about 13% of EMI-2 and 8% of TROPOMI outliers can be screened out.

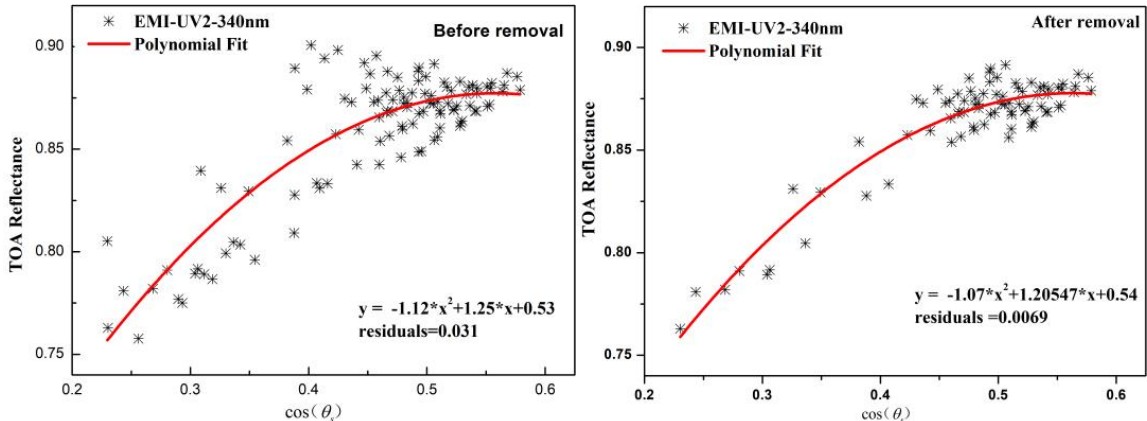

**Figure 5.** *Cont.*

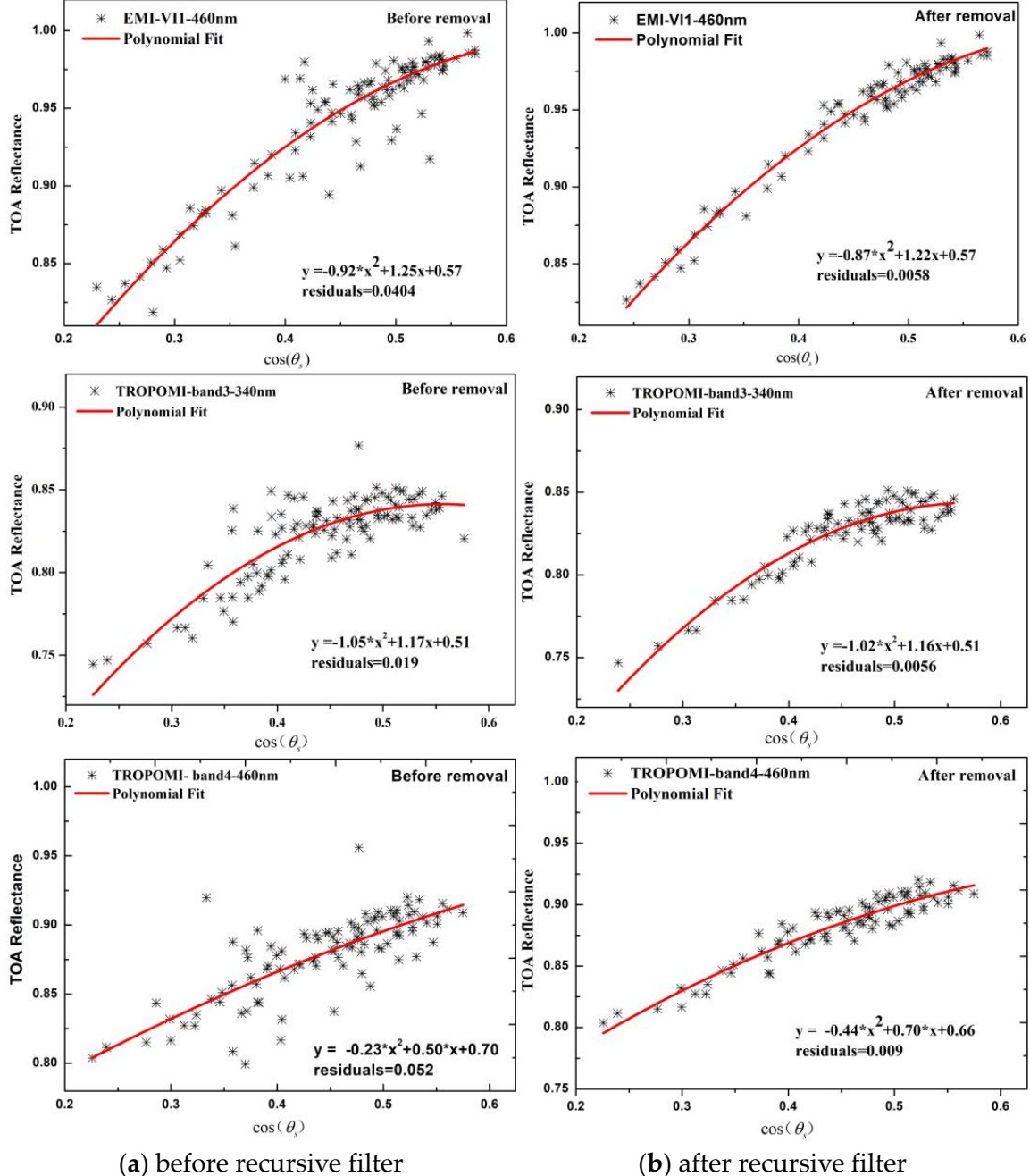

(**a**) before recursive filter        (**b**) after recursive filter

**Figure 5.** Polynomial relationship between the reflectance factor and SZA for EMI-2/TROPOMI data.

### 3.2. TOA Radiance of Reflectance

#### 3.2.1. TOA Reflectance Calculation

Assuming that the top of the atmosphere is a Lambertian reflecting surface, and the sunlight is incident to the surface at the zenith angle $\theta_s$, the TOA reflectance $\rho_{\text{TOA}}(\lambda)$ could be calculated according to the Lambertian reflectance definition, as shown in Equation (2).

$$\rho_{\text{TOA}}(\lambda) = \frac{\pi \cdot L(\lambda) \cdot d^2}{E_s(\lambda) \cdot \cos \theta_s} \tag{2}$$

where $L\left(\mu W \cdot cm^{-2} \cdot sr^{-1} \cdot nm^{-1}\right)$ is the radiance measurement for satellite sensors, $d$ is the Earth-Sun distance in astronomical units, and $E_s(\lambda)\left(\mu W \cdot cm^{-2} \cdot nm^{-1}\right)$ is the measurement of solar irradiance at the TOA for satellite sensors.

The spectral imager data in each channel are related to the instrument spectral response function (ISRF). Spectral channel matching is required for the inconsistency of the channels and spectral response functions (SRF) among different sensors. The SRF of a spectral sensor is generally obtained according to the central wavelength and full width half maximum (*FWHM*). The commonly used SRF is the typical Gaussian response function, expressed as Equation (3).

$$F(\lambda) = \frac{1}{\sqrt{2\pi}\sigma} e^{\frac{-(\lambda-\lambda_i)^2}{2\sigma^2}} \tag{3}$$

$$\sigma = \frac{FWHM}{2\sqrt{2\ln 2}} \tag{4}$$

where $\lambda$ is the wavelength, $\lambda_i$ is the central wavelength of a pixel, and $\sigma$ can be calculated by the *FWHM* as Equation (4). The input TOA radiance and TOA reflectance of the reference sensor can be interpolated with a cubic spline function to share the same channel and spectral response as the sensor to be calibrated.

### 3.2.2. Bands and Wavelengths Selection

In this paper, the TOA radiance of reflectance validation was performed for a selection of two bands with 12 central wavelengths (shown in Figure 6). Figure 6 presents the TOA reflectance measured by EMI-2 over Dome C on 8 November 2021 with no clouds present. The positions of the 12 central wavelengths were indicated with dotted lines. These wavelengths also cover the corresponding spectral range of EMI-2 as much as possible to promise minimal interference from the surrounding absorption bands, which are labeled with corresponding trace gases [3]. Due to the large time span between EMI-2 and TROPOMI calibration, the reflectivity below 325 nm is strongly dependent on the ozone profile [23]. Considering that the variation of Antarctic ozone content with time will greatly affect the accuracy of the calibration, the shortest wavelength listed is set at 328 nm.

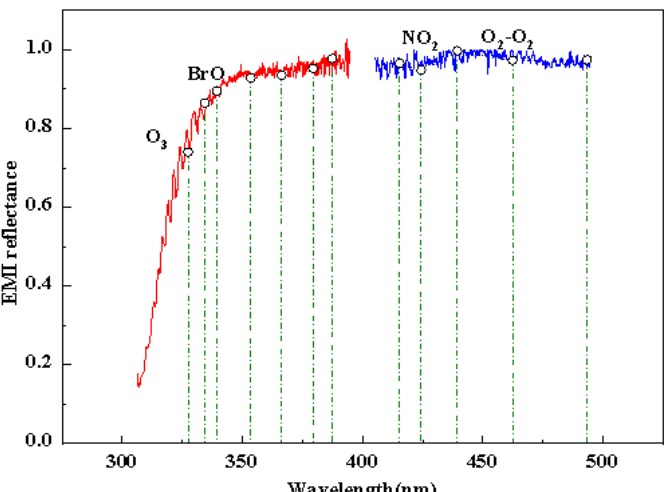

**Figure 6.** TOA reflectance measured by EMI-2 on 8 February 2022 over Dome C.

### 3.3. Spectral Adjustment Factor Correction

Due to the inconsistency of the channels and spectral response functions, two sensors possess different responses to the same radiation source [24], which would lead to the radiation quantity deviation. The cross-calibration of EMI-2 and TROPOMI requires spectral adjustment factor calculation to reduce the cross-calibration uncertainties because of the spectral differences between the analogous bands of the sensors and compensate for the ISRF mismatch [1].

To perform an accurate cross-calibration between these two sensors, the differences due to spectral responses need to be understood and quantified. Because the Sun is

regarded as a relatively stable light source, the measurement of solar irradiance provides a stability reference. Similar to the calculation principle of the SBAF factor [24], the spectral adjustment factor $A_\lambda$ between EMI-2 and TROPOMI can be characterized by the solar spectra measured by itself, as shown in Equation (5).

$$A_\lambda = \frac{E_{EMI}/(S_{refer}(\lambda) \otimes ISRF_{EMI}(\omega,k))}{E_{TROPOMI}/(S_{refer}(\lambda) \otimes ISRF_{TROPOMI}(\omega,k))} \tag{5}$$

where $E_{EMI}$ and $E_{TROPOMI}$ are the solar irradiance measurements of EMI-2 and TROPOMI, respectively, which could be calculated as the right side of the Equation (5). $ISRF_{EMI}(\omega,k)$ and $ISRF_{TROPOMI}(\omega,k)$ represents the instrument spectral response function (ISRF) of EMI-2 and TROPOMI, respectively, and $\lambda$ is the wavelength under specific bandwidth. $S_{refer}(\lambda)$ is the high-resolution solar spectrum from the SAO2010 solar irradiance reference spectrum. Then, the SAO2010 solar irradiance reference spectrum is convoluted with the ISRF of two sensors. After the convolution, the resulting spectra are pixel sampled to match the axis.

*3.4. RTM-Based Correction Factor*

Different viewing geometry, measuring time, meteorological conditions, and spectral response would lead to different measurements of surface radiance [25]. To eliminate the difference due to the atmosphere on the Earth sensor and the difference in view geometry, another correction method is derived in this work using RTM-based, i.e., SCIATRAN, simulations. The radiative transmission modeling-based correction factor to account for the Earth-to-sensor path difference can be represented by a proportionality relation between TOA radiance and direct-ground-reflected radiance.

In this study, the SCIATRAN radiative transfer model is selected to simulate the TOA reflectance of the two sensors, EMI-2 and TROPOMI, under corresponding viewing conditions. The radiative transfer model SCIATRAN developed by the Institute of Environmental Physics, University of Bremen is a comprehensive software package for the modeling of radiative transfer processes in the terrestrial atmosphere and ocean in the spectral range from the ultraviolet to the thermal infrared. The model is capable of simulating transmittance, scattering, and surface reflection processes as well as thermal emission and, thus, is suitable for a wide range of applications related to the remote sensing of the Earth's atmosphere [26,27].

The OMI surface LER database is used for the surface albedo in the SCIATRAN calculation, including the monthly climate database grid with a spatial resolution of $0.5° \times 0.5°$, covering the wavelength range of 328–499 nm [28].

Other input parameters include viewing geometrics, surface height, trace gas concentration from EMI-2 data products, etc. The surface height is from the GMTED2010 (Global Multi-resolution Special Rain Elevation Data) surface elevation database [29].

The correction factor accounting for the Earth-to-sensor path difference can be represented by a ratio between the actual measurements of TOA reflectance and the simulation calculations with direct ground-reflected radiance reflection [11]. The ratio is considered as Equation (6).

$$f_{\lambda,\theta_v,\theta_s,\varphi} = \frac{L_{\lambda,\theta_v,\theta_s,\varphi}}{l_{\lambda,\theta_v,\theta_s,\varphi}} \tag{6}$$

where $f_{\lambda,\theta_v,\theta_s,\varphi}$ is the Earth-to-sensor atmospheric transmission correction factor with given $RAA(\varphi)$, $SZA(\theta_s)$ and $VZA(\theta_V)$ for wavelength $\lambda$, $L_{\lambda,\theta_v,\theta_s,\varphi}$ represents the actual measurements of TOA radiance, while $l_{\lambda,\theta_v,\theta_s,\varphi}$ means the simulation results calculated by SCIATRAN model.

Consequently, the RTM simulation correction factor for the equivalent channels with central wavelength $\lambda$ can be calculated as Equation (7).

$$F_{\lambda,EMI\_TROPO} = \frac{f_{\lambda,\theta_{v-EMI},\theta_{s-EMI},\varphi-EMI}}{f_{\lambda,\theta_{v-TROPO},\theta_{s-TROPO},\varphi-TROPO}} \frac{L_{\lambda,\theta_{v-EMI},\theta_{s-EMI},\varphi-EMI}/l_{\lambda,\theta_{v-EMI},\theta_{s-EMI},\varphi-EMI}}{L_{\lambda,\theta_{v-TROPO},\theta_{s-TROPO},\varphi-TROPO}/l_{\lambda,\theta_{v-TROPO},\theta_{s-TROPO},\varphi-TROPO}},$$

(7)

where $f_{\lambda,\theta_{v-EMI},\theta_{s-EMI},\varphi-EMI}$ and $f_{\lambda,\theta_{v-TROPO},\theta_{s-TROPO},\varphi-TROPO}$ are the Earth-to-sensor path atmospheric correction factors for EMI-2 and TROPOMI, respectively. Parameters $RAA(\varphi)$, $SZA(\theta_s)$ and $VZA(\theta_V)$ represent each matched pair of EMI-2 and TROPOMI observations.

In this study, the TOA reflectance ratio is used as a parameter to evaluate the consistency between sensors, which is demonstrated as Equation (8).

$$C_{i,\lambda} = \frac{\rho_{i,\lambda,EMI}}{\rho_{i,\lambda,TROPO} \times A_\lambda \times F_{i,\lambda,EMI\_TROPO}}$$

(8)

For case $i$ and wavelength $\lambda$, $\rho_{i,\lambda,EMI}$ and $\rho_{i,\lambda,TROPO}$ represents the corresponding at-sensor TOA reflectance measurements. RTM correction factor $F_{i,\lambda,EMI\_TROPO}$ comes from Equation (7), and $A_\lambda$ is the calibration correction adjustment factors derived from Equation (5).

## 4. Results and Discussion

The dataset of this study is selected from the radiance measurement of EMI-2 over the Dome C site from 10 November 2021 to 15 February 2022. Correspondingly, the radiance measured by TROPOMI is synchronously obtained for comparison.

### 4.1. Spectral Adjustment Factors

Figure 7 presents the solar irradiance measured by EMI-2 and TROPOMI on 23 December 2021, respectively, and the high-resolution reference spectrum (SAO2010) is convolved with the EMI-2 ISRF model. The spectral range is 300–400 nm (Ultraviolet) and 400–500 nm (Visible), which contains UV2 and VI1 bands of EMI-2 and Band3 and Band4 bands of TROPOMI.

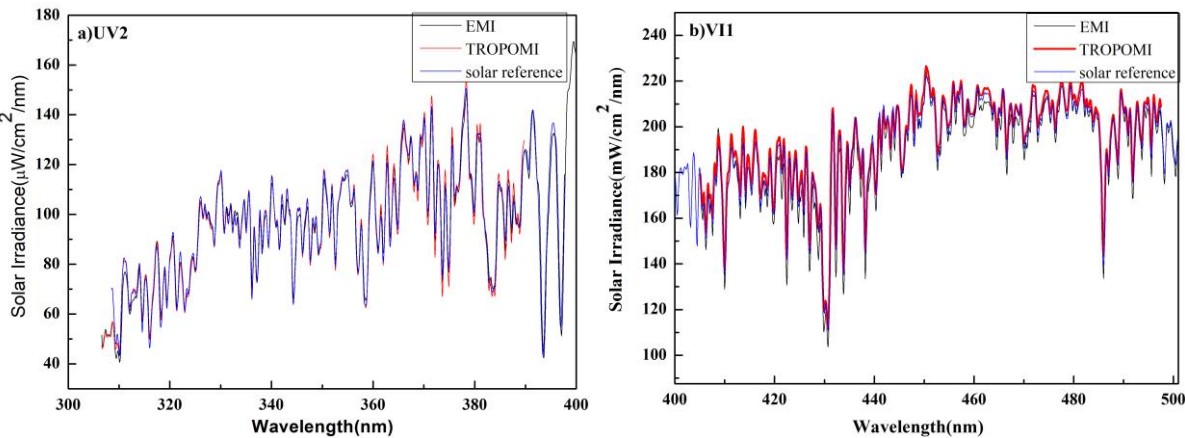

**Figure 7.** The irradiance spectra in ultraviolet (**a**) and visible (**b**) bands measured by EMI-2 (black) and TROPOMI (red) on 23 December 2021. The blue line is from SAO2010 solar irradiance reference spectrum and convolved with the ISRF of EMI-2.

Three solar measurements were from November 2021 to February 2022. Figure 8 shows the results of the spectral adjustment factors (SAFs) for EMI-2 and TROPOMI according to Equation (4). And Table 2 lists the specific values of SAFs and the mean and standard deviation of three cases for EMI-2 and TROPOMI are also calculated. As shown in Table 2,

the mean of SAFs ranges from 0.99 to 1.04; the standard deviation is 0.001~0.002. It can be seen that the largest standard deviation of all the channels is 0.002, which proves that the measurements of the two sensors are relatively stable, and the mean SAFs are applicable for this work.

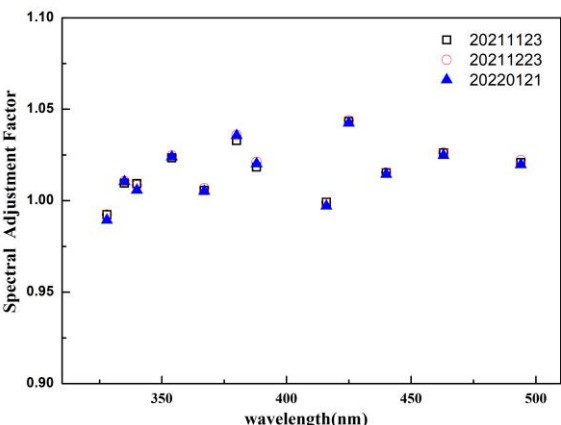

**Figure 8.** Comparison of spectral adjustment factors.

**Table 2.** Spectral adjustment factors for EMI-2 and TROPOMI.

| Wavelength (nm) | 328 | 335 | 340 | 354 | 367 | 380 | 388 | 416 | 425 | 440 | 463 | 494 |
|---|---|---|---|---|---|---|---|---|---|---|---|---|
| 23 November 2021 | 0.992 | 1.010 | 1.009 | 1.023 | 1.006 | 1.033 | 1.018 | 1.000 | 1.043 | 1.015 | 1.026 | 1.021 |
| 23 December 2021 | 0.992 | 1.011 | 1.008 | 1.025 | 1.007 | 1.036 | 1.021 | 0.999 | 1.044 | 1.015 | 1.026 | 1.022 |
| 21 January 2022 | 0.989 | 1.010 | 1.006 | 1.024 | 1.005 | 1.036 | 1.020 | 0.997 | 1.042 | 1.014 | 1.025 | 1.020 |
| Mean | 0.991 | 1.010 | 1.008 | 1.024 | 1.006 | 1.035 | 1.020 | 0.999 | 1.043 | 1.015 | 1.026 | 1.021 |
| STD | 0.002 | 0.001 | 0.002 | 0.001 | 0.001 | 0.002 | 0.001 | 0.002 | 0.001 | 0.001 | 0.001 | 0.001 |

*4.2. Atmospheric Transmission Correction Factor*

The atmospheric transmission correction factor $f_\lambda$ according to Equation (6) is a typical way to eliminate the difference due to the atmosphere on Earth sensor and the difference in view geometry. Figure 9 demonstrates the time series of atmospheric transmission correction factors of EMI-2 (Black) and TROPOMI (Red) relative to theoretical reflectance calculated with SCIATRAN while data filtering and VZA/RAA limitations are involved. It indicates that EMI-2 is deviated slightly heavier than TROPOMI, about 2% to 3%, except for 416 nm, 425 nm, and 440 nm, which demonstrate large biases on the order of 4% or more, and this is inconsistent with the solar irradiance measured by EMI-2 that is larger than that measured by TROPOMI shown in Figure 8. Similar to UV2 bands, VIS bands also suggest consistently lower values responsible for TROPOMI compared to EMI-2. The temporal trends for bias are nearly unchanged.

As shown in Figure 9, at 328 nm, 335 nm, and 340 nm, the atmospheric transmission correction factor series of EMI-2 and TROPOMI almost overlap, but the deviation from the theoretical values obtained by the SCIATRAN model is relatively large (far away from one). At 328 nm, the bias ranges from −0.3 to −0.2, which is much larger than those of 335 nm and 340 nm. This may attribute to the fact that the atmosphere is very sensitive to the influence of the surrounding light transmission in this band (328 nm); the influence of molecular scattering and aerosol on light transmission is easily affected by the Sun and observation geometry. On the other hand, it may be that the radiance signal at this wavelength is a little weak and very sensitive to the changes in the radiation signal [6].

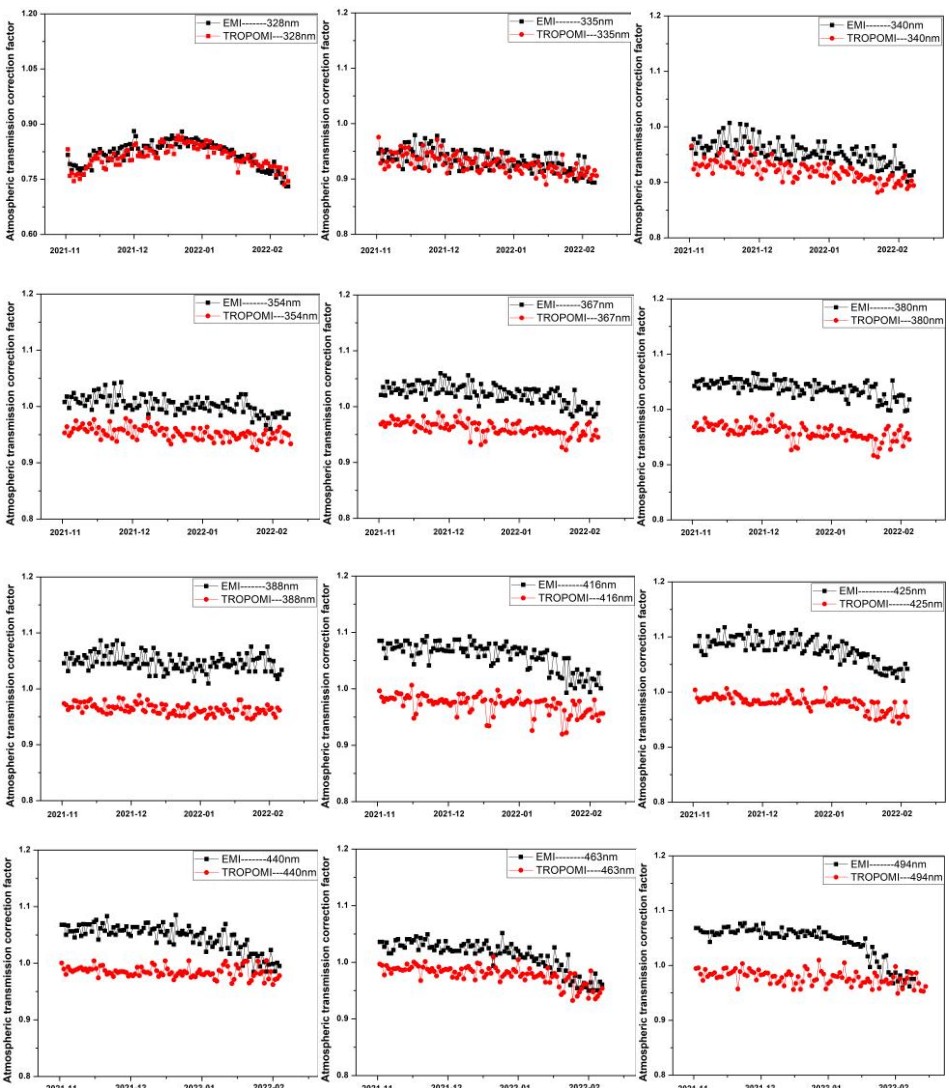

**Figure 9.** Time series of atmospheric transmission correction factor between EMI-2 and TROPOMI for twelve bands.

At 354 nm, 367 nm, 380 nm, and 388 nm, the atmospheric transmission correction factor between measurements and calculations of the SCIATRAN model are close to one, with a small deviation. The ratios between EMI-2 measurements and SCIATRAN model simulations are slightly larger than one (Bias ranges from +0.01~+0.03, and the ratios of measurements and SCIATRAN model simulations are less than one (Bias ranges from −0.05 to −0.02)). In the first three months, the atmospheric transmission correction factor of the TOA reflectance data between the two sensors and the SCIATRAN model deviated heavily, while in February, it became smaller. The reason may be due to the slight sensor degradation of the EMI-2 radiation reflection. It is also not ruled out that a small amount of decay has occurred in the measurement of EMI's orbit to the Earth for three months or because the solar zenith angle at the South Pole in February is mostly greater than 60 degrees, and the radiance values measured by EMI-2 instrument have a greater correlation with the solar zenith angle, which would result in a small radiation reflection.

In the VIS band, the distribution of TROPOMI is closer to one and flatter than those of UV2 bands, and the measurement deviation of EMI-2 is higher than those of UV2 bands, especially in the 416 nm, 425 nm, and 494 nm bands, which may be caused by how the input solar spectrum measured by EMI-2 is larger than that of the TROPOMI at the wavelength during SCIATRAN simulation.

As shown in Figure 10, from November 2021 to February 2022, the monthly deviation of each band is similar. The standard deviation of the atmospheric transmission correction factor ranges from 0.006 to 0.026, and the TOA reflectance is stable. The standard deviations of VI1 bands are smaller than those of UV2 bands. Compared with EMI-2, the standard deviation of TROPOMI is smaller and shows stronger stationary characteristics. In four months, the standard deviation in February is larger. This may be the small amount of data due to the screening conditions requirement and the large zenith angle, so the standard deviation of TOA reflectance measured by the instrument becomes larger.

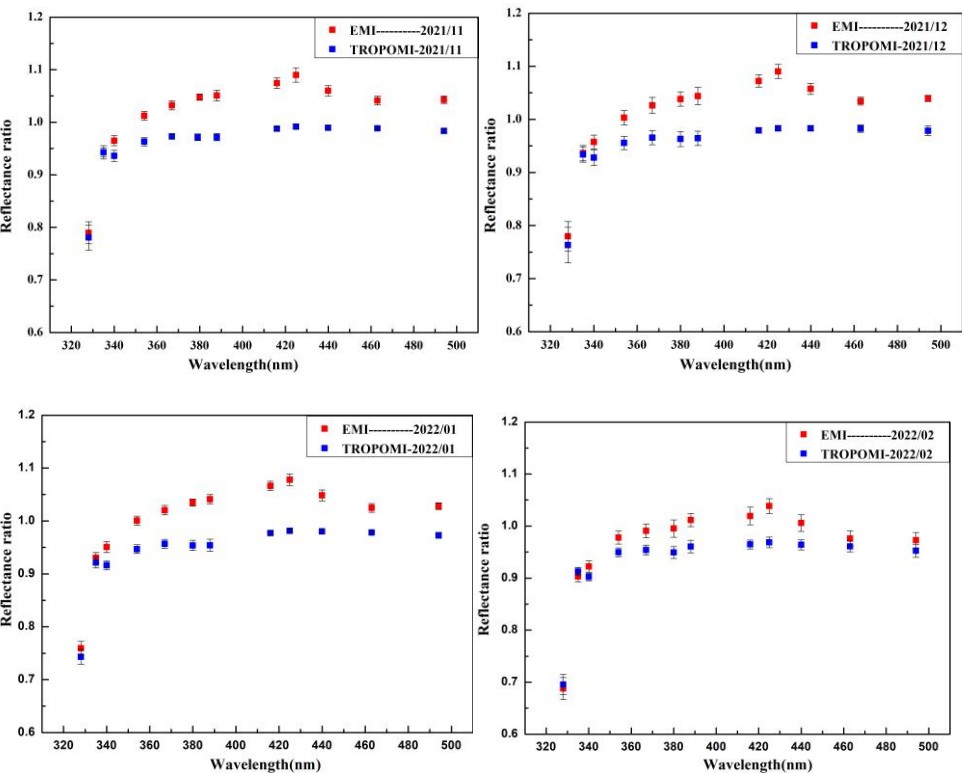

**Figure 10.** Mean and standard deviation series of atmospheric transmission correction factor of EMI-2 and TROPOMI in different months.

As shown in Table 3, except for the 328 nm band and the other individual wavelength, the deviation range of most EMI-2 radiation is within 7%, and the average range of TROPOMI radiation deviation is within 6%.

**Table 3.** The mean and standard deviation of the monthly atmospheric transmission correction factor of EMI and TROPOMI at different wavelengths.

| Wavelength (nm) | EMI-2 | | | | TROPOMI | | | |
|---|---|---|---|---|---|---|---|---|
| | 2021/11 | 2021/12 | 2022/01 | 2022/02 | 2021/11 | 2021/12 | 2022/01 | 2022/02 |
| 328 | 0.79 ± 0.022 | 0.78 ± 0.032 | 0.76 ± 0.013 | 0.69 ± 0.026 | 0.78 ± 0.026 | 0.77 ± 0.033 | 0.74 ± 0.014 | 0.70 ± 0.010 |
| 335 | 0.94 ± 0.012 | 0.93 ± 0.011 | 0.93 ± 0.012 | 0.93 ± 0.017 | 0.94 ± 0.015 | 0.93 ± 0.014 | 0.92 ± 0.011 | 0.91 ± 0.008 |
| 340 | 0.97 ± 0.013 | 0.95 ± 0.018 | 0.95 ± 0.011 | 0.92 ± 0.017 | 0.93 ± 0.013 | 0.93 ± 0.012 | 0.91 ± 0.010 | 0.90 ± 0.008 |
| 354 | 1.01 ± 0.010 | 1.00 ± 0.014 | 1.00 ± 0.012 | 0.97 ± 0.010 | 0.96 ± 0.009 | 0.96 ± 0.011 | 0.95 ± 0.008 | 0.95 ± 0.010 |
| 367 | 1.03 ± 0.009 | 1.03 ± 0.013 | 1.02 ± 0.009 | 0.99 ± 0.014 | 0.97 ± 0.007 | 0.97 ± 0.010 | 0.96 ± 0.007 | 0.95 ± 0.011 |
| 380 | 1.04 ± 0.006 | 1.04 ± 0.011 | 1.03 ± 0.010 | 0.99 ± 0.016 | 0.97 ± 0.007 | 0.96 ± 0.010 | 0.95 ± 0.009 | 0.95 ± 0.012 |
| 388 | 1.05 ± 0.011 | 1.07 ± 0.021 | 1.04 ± 0.012 | 1.01 ± 0.020 | 0.97 ± 0.007 | 0.97 ± 0.009 | 0.96 ± 0.007 | 0.96 ± 0.012 |
| 416 | 1.07 ± 0.012 | 1.09 ± 0.016 | 1.06 ± 0.022 | 1.01 ± 0.013 | 0.99 ± 0.009 | 0.98 ± 0.008 | 0.97 ± 0.014 | 0.97 ± 0.012 |
| 425 | 1.08 ± 0.014 | 1.05 ± 0.016 | 1.07 ± 0.023 | 1.03 ± 0.011 | 0.99 ± 0.006 | 0.98 ± 0.007 | 0.98 ± 0.016 | 0.96 ± 0.014 |
| 440 | 1.05 ± 0.013 | 1.03 ± 0.015 | 1.04 ± 0.024 | 1.00 ± 0.014 | 0.99 ± 0.007 | 0.98 ± 0.008 | 0.98 ± 0.013 | 0.96 ± 0.013 |
| 463 | 1.04 ± 0.009 | 1.03 ± 0.014 | 1.02 ± 0.024 | 0.98 ± 0.010 | 0.99 ± 0.006 | 0.98 ± 0.009 | 0.97 ± 0.013 | 0.96 ± 0.017 |
| 494 | 1.04 ± 0.007 | 1.03 ± 0.009 | 1.03 ± 0.027 | 0.97 ± 0.011 | 0.98 ± 0.007 | 0.97 ± 0.013 | 0.97 ± 0.015 | 0.95 ± 0.019 |

### 4.3. Intercomparison and Cross-Calibration Validation

After the spectral adjustment factor and RTM correction, the time series of the EMI-TROPOMI radiance reflection ratio is derived according to Equation (8), and the cross-calibration calculation of EMI to TROPOMI is implemented.

Figure 11 shows the time series of the reflectance ratio of EMI-2 to TROPOMI with different bands (328~494 nm). The reflectance ratios are fitted with a linear function. From November 2021 to February 2022, the time series change of each wavelength band is nearly linear. The largest slope ($-2 \times 10^{-4}$) lies at 416 nm, and the smallest slope ($-0.76 \times 10^{-5}$) is 335 nm. However, both the largest and the smallest are close to 0, which shows that EMI-2 has a strong correlation with TROPOMI in different wavelength bands. In fact, it also shows that the RTM-based correction method can effectively correct the annual periodic change of the reflectance ratio caused by the observed geometric changes. After RTM-based correction, the reflectance ratio is relatively flatter, but the reflectance data still has a certain angular correlation.

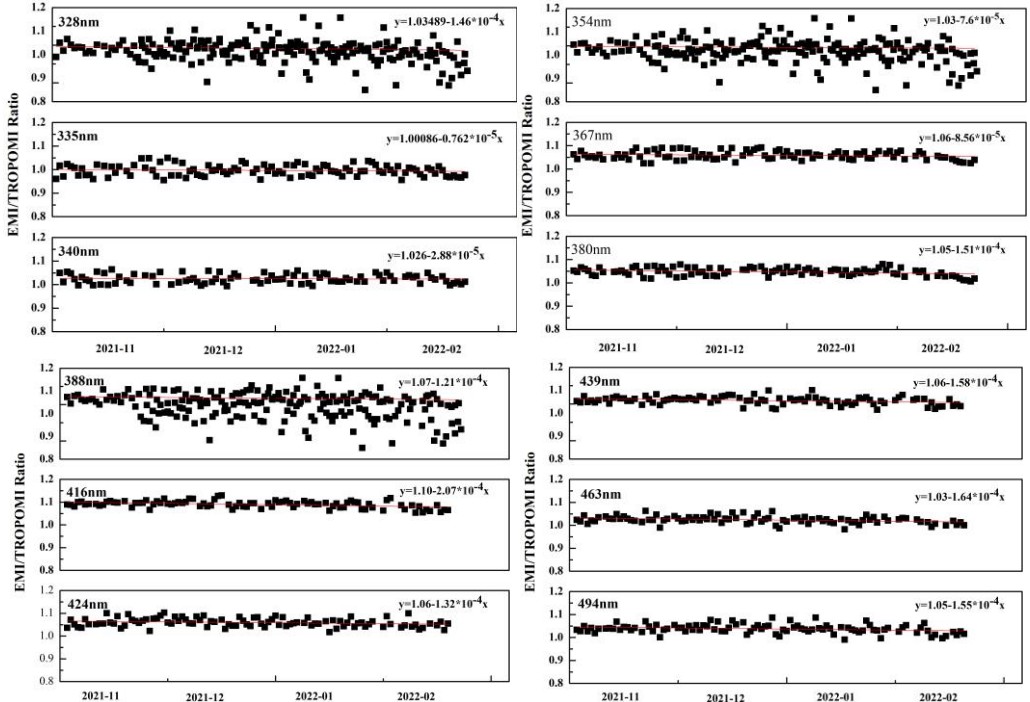

**Figure 11.** Time series of reflectance ratios of EMI-2 to TROPOMI after spectral adjustment factor and RTM-based correction.

Figure 12 shows the mean and standard deviation of the monthly reflectance ratio of EMI-2 to TROPOMI after spectral adjustment factor and RTM correction. The standard deviation of normalized TOA reflectance represents the uncertainty of radiometric calibration, and the change in mean value represents the stability of radiation response. As shown in Figure 12, the standard deviation is less than 3%, which conforms to the variation scope of Antarctic ice and snow reflectivity, and the instrument radiance signal is stable [13]. The reflectance ratios of EMI-2 to TROPOMI range from 0.998 to 1.09. For 328 nm, 335 nm, 340 nm, 460 nm, and 490 nm, the ratios are all close to one, and the biases are less than 3%. For the rest wavelengths, the biases are less than 6%. However, the reflectance ratio of 416 nm is higher than the expected value (deviation < 1.07). In total, the two sensors demonstrate good consistency and strong correlation, which fully meets the requirement that the relative radiometric calibration is less than 7%.

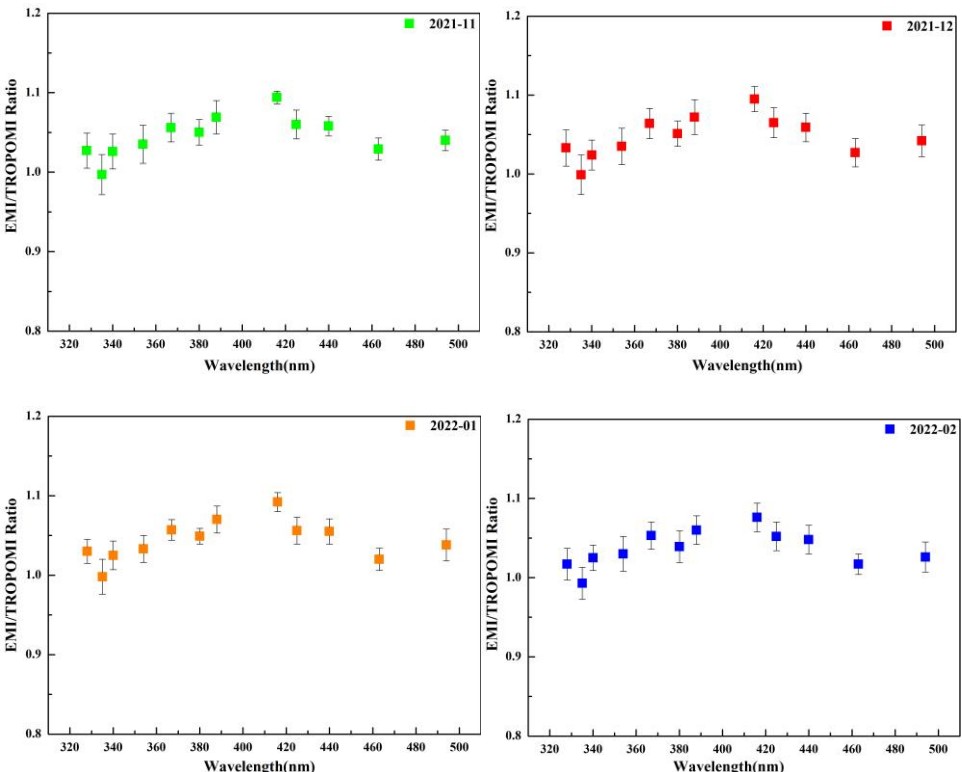

**Figure 12.** The mean and standard deviation of reflectance ratio between EMI-2 and TROPOMI after spectral adjustment factor and RTM correction.

From the perspective of time trend, the relative calibration error of EMI-2 to TROPOMI gradually declined in February 2022. This may attribute to the large solar zenith angle at the time junction of the South and North Pole, the dependence of the residual solar zenith angle, the degradation of EMI-2 radiance value caused by the change of cloud cover and seasonality [12], and also may be caused by a small amount of decay of the instruments after in orbit. As a result, it is necessary and meaningful to continuously track the cross-calibration of EMI-2 and TROPOMI.

## 5. Conclusions

In this research, the radiometric calibration quality of the EMI-2 instrument was investigated by calculating the TOA reflectance data of EMI and TROPOMI obtained from the pseudo-invariant calibration site Dome C. The EMI-2 and the reference instrument TROPOMI were also cross-calibrated, combining SAF and RTM-based correction; the measurements of EMI-2 and TROPOMI are strongly correlated in the UV2 band and VI1 band. For 328 nm, 335 nm, 340 nm, 460 nm, and 490 nm, the ratios are all close to one, and the biases are less than 3%. For the rest wavelengths, the biases are less than 6%. In the first three months, the ratio of the TOA reflectance data between the two sensors deviated heavily, and it decreased significantly in February. Therefore, the radiometric calibration of EMI-2 is relatively accurate and meets the requirements of less than 7% of the average design relative radiometric calibration error, which can be used to maintain the quantitative application of satellite sensors and provide a potentially useful reference for the radiometric cross-calibration of other similar satellite sensors.

**Author Contributions:** Conceptualization, J.S. and F.S.; methodology, J.S. and M.Z.; software, Y.H. and H.Z.; Writing and review, J.S. and F.S. All authors have read and agreed to the published version of the manuscript.

**Funding:** This work was supported by the National Natural Science Foundation of China under Grant 41705016.

**Data Availability Statement:** The data that support the findings of this study are available from the corresponding author, [author initials], upon reasonable request.

**Conflicts of Interest:** The authors declare no conflict of interest.

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
