# Peer review of "Validation of EMI-2 Radiometric Performance with TROPOMI over Dome C Site in Antarctica"

_remotesensing, doi:10.3390/rs15082012_

Round 1

Reviewer 1 Report

Major General Comments

 I have some concerns on the scientific soundness of the applied cross-calibration approach. More specifically the formula used for the Spectral Band Adjustment Factor (SBAF) is not consistent with other literature references (e.g.  Chandler et al. (2012)). I don’t understand why the actual irradiance measured by the sensors is included in the formula. This is not in line with how the SBAF is normally defined. The SBAF is a correction “factor” to correct for differences in spectral bands.   The authors should provide some very good references for the use of the formula that they have proposed.

 It is very confusing that in the results section and in the figures the authors use always “reflectance ratio” for both the results of equation 7 and equation 8. It is not clear for the reader what is exactly given in the figures and how to interpret the results.

Specific Comments

-          Geometric effect/geometric correction (eg. a correction method based on radiative transfer model (RTM) was used to suppress geometric effect; RTM-based method to perform geometric correction of the measurement,…) : radiative transfer models are used for ‘radiometric’ modelling. I don’t understand what is meant with geometric correction.

-          Abstract line 24 ‘SCIATRAN’ is not introduced

-          Abstract line 29: what is meant with “offsets”? The offset of a linear regression?

-          Line 51: How are the solar irradiance measurements performed? Using a solar diffuser N

-          Line 61 to Line 76: is this one sentence? Difficult to follow. Please rewrite this section.

-          P 3: line spacing changes

-          Line 112 range from 240 to 710 nm. I assume this is not for all channels this range. This is not clear.

-          Line 115: what are “airborne” diffusers?

-          3.1.1 angular limitations: why no limitation on the SZA?

Reviewer 2 Report

This paper describes a radiometric validation of EMI-2 through an inter-comparison with TROPOMI over Dome C. Please address the following comments:

General:

Placing the comparisons made in the context of the expected radiometric uncertainties of each instrument would strengthen the analysis. Then we could see if the differences found are expected given their uncertainties.

Specific:

31 “is reliable” is not descriptive. It would be better to say something like within the relative accuracy requirement or within the expected uncertainty.

42 “The retrieval codes of cloud and aerosol products depend on accurate radiometric calibration.” A reference may be needed to support this statement or further description.

63 I am not sure why you are suddenly describing TROPOMI validation. Are you establishing that TROPOMI is valid reference instrument because its calibration has been validated in many ways? If that is the case, you may also reference its SI-traceable pre-launch calibration. (https://amt.copernicus.org/articles/11/6439/2018/#&gid=1&pid=1)

63 “studying the quality of the radiometric calibration” What does this mean?

69 "inner set of on-board calibrators". Can you explain further?

I recommend adding a schematic of the optical path from the sun and Earth and on-board calibrators within the instrument, so that the optical paths for the solar port and Earth port are clear.

103 Reference 5 suggests 2% non-uniformity. I am not sure this necessarily translates to 2% radiance uncertainty for this inter-comparison, since they are lower spatial resolution. What is the area of the site used here?

115 “airborne”. I think you mean “on-board”

130 How accurate and stable is TROMPOMI?

3.1 It sounds like you are trying to say that matching the geometry of the two sensors is important because of the the high latitude of the calibration site, the large viewing angles, and the high zenith angles, create a high TOA reflectance sensitivity to BRDF and Rayleigh effects. Is that correct? Perhaps this description could be clarified.

168 typo “o”

174 What reflectance range is removed here? What value corresponds to the 2-sigma above the average?

I am not sure that using the polynomial fit and removing outliers should be characterized as “recursive” How do you know these outliers are due to cloud shadow effects?

191 “pollution”. You may mean “contaminated”

209 “Lambertian” not “Lambert”

269 Are aerosol and water vapor concentrations used in the radiative transfer model?

316 What does the solar irradiance measurement comparison have to do with the stability of the DOME C site? (Maybe I’m missing something here)

Fig. 8 Is this the ratio between the sensors and the RTM prediction or the comparison between EMI and TROPOMI radiances accounting for the correction factors (SBAF, transmission differences)

 Table 4.3 isn’t necessary. It has the same information as in the plots.

358 “decline of EMI-2 radiation reflection” – Do you mean sensor degradation?

399 “dependence” instead of “correlation” seems more appropriate here.

422 Is the requirement for relative accuracy (defined with respect to a reference sensor) or is it an absolute uncertainty requirement? What is the uncertainty expected for EMI-2 based on the pre-launch measurements and uncertainty of the calibration updates based on on-board calibrators? (I’ve also mentioned this in the general comments section)

Table 5 may not be needed (or included as an appendix). The same information is in Fig. 11.

Round 2

Reviewer 1 Report

The authors have satisfactorily addressed most of my concerns. 

I still would suggest that the authors use another name for the "Spectral band adjustment factor" as normally the SBAF is just the correction factor, while in equation 5 the authors already apply it to the EEMI  / ETROPOMI ratio. 
